# Detectability of *Pseudomonas syringae* pv. *aesculi* from European Horse Chestnut Using Quantitative PCR Compared with Traditional Isolation

**Salome Schneider** [1,*], **Christopher Schefer** [1] **and Joana Beatrice Meyer** [1,2]

1   Forest Health and Biotic Interaction, Swiss Federal Institute for Forest, Snow, and Landscape Research WSL, 8903 Birmensdorf, Switzerland; christopher.schefer@hotmail.com (C.S.); joana.meyer@bafu.admin.ch (J.B.M.)
2   Forest Division, Federal Office for the Environment FOEN, 3063 Ittigen, Switzerland
*   Correspondence: salome.schneider@wsl.ch; Tel.: +41-44-739-23-47

**Abstract:** Bleeding cankers on horse chestnut trees (*Aesculus hippocastanum* and *Aesculus × carnea*), caused by *Pseudomonas syringae* pv. *aesculi*, have been reported across Europe. In the present study, we show the successful detection of *P. syringae* pv. *aesculi* on symptomatic horse chestnut trees in Switzerland using quantitative PCR (qPCR). However, *P. syringae* pv. *aesculi* was also detected by qPCR on trees from which no isolate was obtained through cultivation. Reduced isolation success and low copy numbers of the target gene were correlated with the increasing age of symptomatic horse chestnut trees. The potential of detecting non-viable *P. syringae* pv. *aesculi* by qPCR was evaluated using an inoculation experiment with dead bacteria and detection by qPCR and cultivation. The detectability of DNA from *P. syringae* pv. *aesculi* cells dropped by 34.5% one day after inoculation and then decreased only slightly until the end of the experiment (22 days after inoculation). In contrast, no bacterial growth was observed at any time point after the inactivation of the bacteria. To protect horse chestnut trees, evaluating the viability and actual infection stage of the bacterium may play an important role.

**Keywords:** *Aesculus hippocastanum*; bacterium; bleeding canker; molecular diagnostic tool; *Pseudomonas syringae* pv. *aesculi*; real-time PCR; inactive bacteria

## 1. Introduction

Bleeding cankers on European horse chestnut (*Aesculus hippocastanum*) trees were observed for the first time in the early 1970s in the southern UK [1]. Since then, the incidence has largely increased in the UK and the disease has also been reported in many other countries in Europe, including Ireland, The Netherlands, Belgium, Czech Republic, Slovenia, Germany, and Switzerland [2–10]. The bacterium *Pseudomonas syringae* pv. *aesculi* has been identified as the main causing agent of bleeding cankers [11]. This bacterium was first discovered in 1980 in India, causing minor leaf spots on the Indian horse chestnut (*Aesculus indica*) [12]. Early symptoms include brownish colored spots and patchy bleeding lesions on the trunk or branches. This can progress further and lead to increasing the death of phloem and cambium tissues, so that the water and nutrient transport are impaired, and individual branches or even the entire tree may die [13]. In Europe, this bacterium has spread fast on new susceptible host species and in the UK, for instance, it is estimated to be present on ~50% of horse chestnut trees [14]. In addition to the white horse chestnut (*A. hippocastanum*), the red horse chestnut (*Aesculus × carnea*) is also affected by the disease [3].

Isolation of *P. syringae* pv. *aesculi* from infected host tissue is challenging, time consuming, and therefore expensive. In diagnostics, quantitative PCR (qPCR) is widely used for the rapid and reliable detection of *P. syringae* pv. *aesculi* in host tissue [4,8]. Because of its sensitivity, even low pathogen loads can be detected and measurements can be undertaken

to avoid further spreading and protect surrounding trees. Nevertheless, one advantage of cultivation-based detection is that the presence of viable bacteria is confirmed, which have a potential to multiply and further weaken the tree, as well as infect other trees. In contrast, using qPCR, also non-viable bacteria can be detected [15,16].

Therefore, the aim of this study was to assess symptomatic trees for the presence of *P. syringae* pv. *aesculi* at different locations in Switzerland using qPCR and cultivation-based isolation of the bacterium. Young trees that had been planted recently, as well older trees that had been planted six or more years ago and were showing symptoms, were included. The potential of detecting non-viable *P. syringae* pv. *aesculi* on stem samples by qPCR was further assessed in a time course experiment using chestnut stems inoculated with non-viable bacteria. This study may contribute to a better interpretation of qPCR data regarding the infection stage of bleeding cankers on European chestnut trees, helping to evaluate disease progress and the infection risk for horse chestnut trees in the surroundings.

## 2. Materials and Methods

### 2.1. Sites

The study was conducted at three locations in Switzerland, Zurich, Klingnau, and Basel (Table 1). At each location, bleeding cankers were observed on horse chestnut trees (*A. hippocastanum* and *Aesculus × carnea*). In Zurich, bleeding cankers were present on freshly planted trees, except for one which was found on an older tree within an old population of horse chestnut trees. In Klingnau, only young trees that were freshly planted along a road were affected. In Basel, the sampled trees had been planted at least six years prior in a park with older horse chestnut trees.

**Table 1.** Detection of *Pseudomonas syringae* pv. *aesculi* (PSA) in bark and exudate samples collected from bleeding lesions on *Aescculus* spp. by quantitative PCR (qPCR) and isolation. A tree was considered to be infected by PSA if at least one sample tested positive for PSA by qPCR or isolation.

| Location | Tree | Tree Species | Tree Age (Yrs) | Bleeding Lesions | Sample | qPCR | Isolation |
|---|---|---|---|---|---|---|---|
| Zurich | Tree 1 | *A. hypocastanum* | 1 | Many lesions at the stem | Bark | 3 (3)[1] | 2 (4)[1] |
| - | - | - | - | - | Exudate | 0 (1) | 0 (2) |
| - | Tree 2 | *A. hypocastanum* | 1 | Many lesions at the stem | Bark | 3 (3) | 2 (4) |
| - | - | - | - | - | Exudate | 1 (1) | 0 (2) |
| - | Tree 3 | *A. hypocastanum* | 1 | Many lesions at the stem | Bark | 3 (3) | 1 (4) |
| - | - | - | - | - | Exudate | 1 (1) | 2 (2) |
| - | Tree 4 | *A. hypocastanum* | 1 | Many lesions at the stem | Bark | 3 (3) | 0 (4) |
| - | - | - | - | - | Exudate | 1 (1) | 0 (2) |
| - | Tree 5 | *A. hypocastanum* | >10 | One lesion on one branch | Bark | 0 (3) | 0 (4) |
| - | - | - | - | - | Exudate | 1 (1) | 0 (2) |
| Basel | Tree 1 | *A. × carnea* "Briotii" | 6 | One lesion at the stem | Bark | 1 (3) | 0 (4) |
| - | Tree 2 | *A. × carnea* "Briotii" | 6 | One lesion at the stem | Bark | 3 (3) | 0 (4) |
| - | Tree 3 | *A. hypocastanum* | 33 | Many lesions at the stem | Bark | 1 (3) | 0 (4) |
| - | Tree 4 | *A. × carnea* | 63 | One lesion at the stem | Bark | 2 (3) | 0 (4) |
| - | Tree 5 | *A. × carnea* | 23 | Many lesions at the stem | Bark | 0 (3) | 0 (4) |
| Klingnau | Tree 1 | *A. × carnea* | 3 | Many lesions at the stem | Bark | 3 (3) | 0 (4) |
| - | Tree 2 | *A. hypocastanum* | 3 | Many lesions at the stem | Bark | 3 (3) | 0 (4) |
| - | Tree 3 | *A. × carnea* | 3 | Many lesions at the stem | Bark | 3 (3) | 1 (4) |
| - | Tree 4 | *A. hypocastanum* | 3 | Many lesions at the stem, no exudate | Bark | 3 (3) | 0 (4) |

[1] Number of positive samples (total number of samples analyzed).

## 2.2. Samples

From each tree, bark samples were collected from bleeding lesions using a sterilized JamshidiTM Crown Bone Marrow Biopsy borer (Care Fusion, San Diego, CA, USA), 2 mm in diameter. In Zurich, samples were also taken from the exudates of the bleeding lesions with a sterilized toothpick.

## 2.3. Cultivation-Dependent Detection of P. syringae pv. aesculi

For isolation of the bacteria, bark and exudate samples from each tree were randomly chosen and incubated without surface sterilization on nutrient agar (N9405; Fluka, Buchs, Switzerland) amended with 5% D(+) sucrose (84,100; Fluka), 2 mg/L crystal violet (C0772; Sigma-Aldrich, Buchs, Switzerland) and 50 mg/L cycloheximide (A0879,0005; AppliChem, Darmstadt, Germany). Fluorescent bacteria were isolated and maintained on King's B medium (84,100; Fluka) amended with 10 mL/L glycerol solution (4978; Sigma-Aldrich) and 50 mg/L cycloheximide. For DNA extraction, bacterial cells were suspended in sterile water and lysed through boiling. DNA-containing extracts were separated from cell debris by centrifugation and for species identification, *P. syringae* pv. *aesculi* specific qPCR was used as described in Section 2.4.

## 2.4. Cultivation-Independent Detection of P. syringae pv. aesculi

The total DNA was extracted from randomly chosen bark and exudate samples on sterile toothpicks from each tree, as described by Schneider et al. [17]. Quantitative PCR was conducted in a reaction volume of 20 µL, containing 5 µL of 1:10 diluted DNA from the bark or exudate samples, using 1 × GoTaq qPCR Master Mix (Promega, Fitchburg, WI, USA), and 0.2 mg/mL Bovine Serum Albumin (Sigma-Aldrich). The primers AM-Aes1F and AM-Aes1R and cycling conditions corresponded to the protocol described by McEvoy et al. [4]. Following amplification, a melting curve analysis was performed to confirm the expected product size, with temperatures ranging from 60 to 95 °C and continuous fluorescence measurements taken at every 0.1 °C/s increase in temperature. For each sample, five technical replicates were run in an ABI 7500 Fast Real-Time System (Applied Biosystems, Rotkreuz, Switzerland). The threshold line was manually set in the exponential phase of the amplification curve for determining the sample-specific threshold cycle numbers (CT). As a quantification standard in the qPCR assays, a 10-fold serial dilution from $5 \times 10^7$ to 5 copies/µL of a plasmid was used, containing the DNA sequence obtained from the AM-Aes1 primer pair of the reference culture *P. syringae* pv. *aesculi* CCOS951 [10]. By adding 5 µL of the quantification standard to the reaction mix, 25 copies was the lowest point in the quantification standard, representing the technical detection limit of this assay. Efficiency values and correlation coefficients were calculated for each standard quantification curve of CT against the logarithm of the number of input copies of the target. Samples analyzed for the presence of *P. syringae* pv. *aesculi* through qPCR were considered positive if three technical replicates were above the detection limit.

## 2.5. Detection of Non-Viable P. syringae pv. aesculi by qPCR

A bacterial suspension from the reference culture *P. syringae* pv. *aesculi* CCOS951 was boiled three times for 20 s using a Bunsen burner. Mortality was checked by incubating 50 µL of 1:10, 1:100, and 1:1000 dilutions in triplicate on LB Agar plates before and directly after the heat treatment, and thr numbers of colony forming units (CFU) were counted. Aliquots of each dilution step were taken for DNA extraction, followed by qPCR for quantification, as described above. Stem samples of 0.7 cm length from five living horse chestnut seedlings (2 years old) collected in early spring were completely covered by the heat-treated bacteria solution and incubated at room temperature for 4 h in a Falcon tube. The stem samples were then removed from the solution and put individually in wells of a titer plate, covered with an AirPore tape sheet (Qiagen), and stored at room temperature in the dark. Triplicate samples of each tree were taken daily for 22 days and stored at −20 °C. In addition, one stem sample incubated with sterile water was collected daily, which served

as the negative control. DNA extraction of stem samples and quantification of the number of detectable gene copies by qPCR using the AM-Aes1 primer pair and standard curve calculation were done as described above. A sample time point was scored a positive result if at least two replicates were above the detection threshold.

### 2.6. Statistical Analyses

For determination of the significant differences of the *P. syringae* pv. *aesculi* gene copy numbers between the bark and exudate samples in Zurich, the unpaired two-samples Wilcoxon rank sum test implemented in the R package "dplyr" [18] run with RStudio [19] was used. Correlations between *P. syringae* pv. *aesculi* gene copy numbers and successful isolation of the bacterium, as well as tree age, were calculated using the Spearman correlation test implemented in the R package "stats" [20]. The detection of DNA from non-viable bacteria over time was tested with a linear fixed-effects model "lmer" in the R package "lme4" [21]. The strongest model fit was found for the model y~tree + day + (1 | tree) based on the AIC score. Significant differences between trees and the sampling date were investigated by Tukey HSD using the function "glht". Figures were generated with the package "ggplot2" [22] in R.

## 3. Results

A total of 14 symptomatic trees from three different locations in Switzerland were investigated for the presence of *Pseudomonas syringae* pv. *aesculi* by qPCR and cultivation (Table 1).

### 3.1. Cultivation-Independent Detection of P. syringae pv. aesculi

Using qPCR, all of the trees tested were positive for *P. syringae* pv. *aesculin,* except from one tree located in Basel (Table 1 and Table S1). In Zurich, both the bark and exudate samples were collected from each tree, whereas in Basel and Klingnau, only the bark samples were taken. The number of gene copies from the bacterium detected in the exudate and bark samples in Zurich varied between 5.3 and $3.0 \times 10^6$, and a significantly higher number of gene copies was found in the bark compared with the exudate samples ($W = 61$, $p = 0.045$). A significant negative correlation ($r = -0.72$, $p = 0.0037$) was found between the number of gene copies and tree age. All of the trees that were six or more years old were located in Basel, with the exception of one tree that was located in Zurich. Clear symptoms were found, however, detection by qPCR was below 200 copies/reaction. The number of gene copies in the bark samples from trees younger than six years was at least $2.8 \times 10^5$. The only exception was found in Klingnau, where $1.9 \times 10^3$ ($\pm 1.5 \times 10^3$) gene copies were found in the bark samples from a tree showing only lesions without exudates (Table 1).

### 3.2. Cultivation-Dependent Detection of P. syringae pv. aesculi

The bacterium was isolated from four trees, of which three were located in Zurich. The number of *P. syringae* pv. *aesculi* gene copies detected in the bark samples from these three trees was above $4.2 \times 10^5$ ($\pm 3.4 \times 10^5$) in all of the samples. The fourth tree was located in Klingnau and showed the highest number of gene copies ($1.8 \times 10^6 \pm 7.9 \times 10^5$) found in this study. Successful isolation of *P. syringae* pv. *aesculi* significantly correlated ($r = 0.59$ and $p = 0.0079$) with the increasing number of gene copies detected by qPCR. Isolation of the bacterium from the exudate was successful for only one sample with $1.7 \times 10^3$ copies of the target gene. No *P. syringae* pv. *aesculi* was isolated from the trees that were six years old or older.

### 3.3. Detection of Non-Viable P. syringae pv. aesculi by qPCR

The detectability of DNA from non-viable *P. syringae* pv. *aesculi* cells on wood samples over a period of 22 days was tested. Cultivation of the bacterium before the heat treatment resulted in $3.2 \times 10^6$ ($\pm 8.9 \times 10^5$) CFU/µL, whereas after heat treatment, no bacterial growth was observed. Using qPCR, $2.7 \times 10^7$ ($\pm 5.9 \times 10^6$) and $3.1 \times 10^7$ ($\pm 3.8 \times 10^6$)

copies/µL of the target gene from *P. syringae* pv. *aesculi* were detected before and right after the heat treatment, respectively. From the first to the second day after inoculation, a drop in gene copy numbers detected on the stem samples was observed (34.5% ± 2.4; Figure 1 and Table S2). A reduced detectability of the target DNA over time was observed for all sample sets; however, the decrease in detectable bacteria over the sampling time was significant for only three subsets (Figure 1 and Table S2). In addition, significant differences in the detectability of the bacterium over time were found between the trees (Figure 1). More than three weeks after the heat treatment, between $4.5 \times 10^6$ and $2.1 \times 10^7$ copy numbers were detected on the stem samples. Even 33 months after heat treatment, $2.8 \times 10^5$ ($\pm 1.9 \times 10^5$) copies were found on the stem samples, although no bacterial growth was observed.

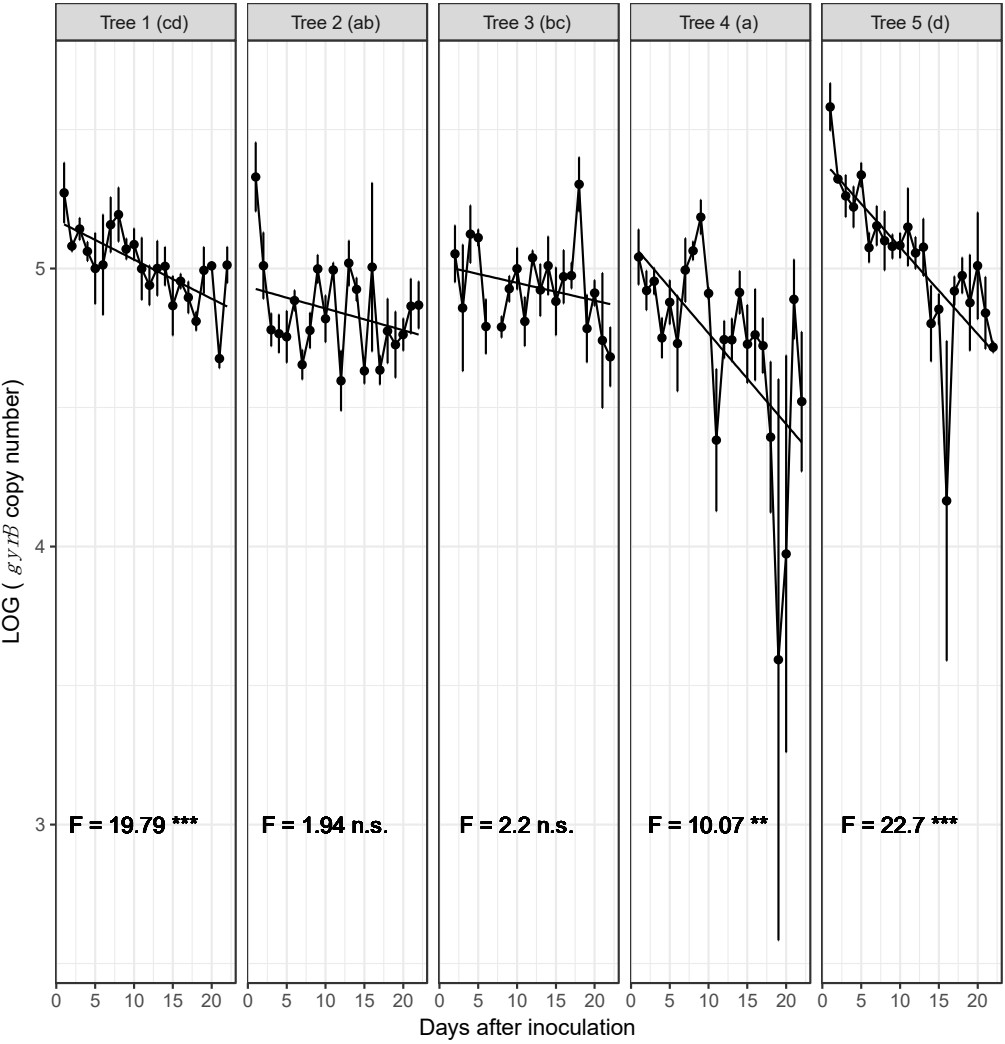

**Figure 1.** Gene copy numbers of the *gyrase B* (gyrB) gene targeted for daily detection of non-viable *Pseudomonas syringae* pv. *aesculi* cells on stem samples collected from five different trees over an incubation period of 22 days. Stem samples were collected for DNA extraction and quantitative PCR each day after inoculation with non-viable *P. syringae* pv. *aesculi* cells. F-values and significance (*** $p < 0.001$, ** $p < 0.01$, n.s. not significant) of the number of days after inoculation based on a linear fixed-effect model y~tree + day + (1 | tree) are presented for each tree. For each tree, a smoothing line based on the linear model according to the number of days after inoculation has been added. Trees with different lower-case letters differ significantly in *P. syringae* pv. *aesculi* abundance.

## 4. Discussion

The presence of the bacterium *P. syringae* pv. *aesculi* could be confirmed at all locations on all horse chestnut trees showing bleeding cankers analyzed in this study. This finding confirms a wider distribution of this pathogen in Switzerland than was expected after its first detection in 2015 [10]. A similar development was observed in Germany, where the bacterium was first found in Hamburg through isolation [6], and a wider distribution was confirmed in later studies using molecular tools [23,24]. Taken together, these studies nicely demonstrate that the monitoring of this particular pathogen can be improved with cultivation-independent detection tools. Similar cultivation-independent diagnostic tools are available for other microorganisms and are used for large-scale monitoring in different ecosystems [17,25,26].

In the present study, horse chestnut trees with bleeding cankers were analyzed that were recently planted along a road forming a newly established alley (Klingnau) or in a park with old horse chestnut trees (Zurich), as well as trees that were planted in parks more than six years ago (Zurich and Basel). The bacteria found on the newly planted trees in Klingnau most likely came with the trees from the nursery, as no other horse chestnut trees were present in the closer surroundings. Similarly, in Zurich, *P. syringae* pv. *aesculi* was found on several recently planted horse chestnut trees. Moreover, the bacterium was also detected on an older horse chestnut tree that was next to an infected, planted tree. This strongly suggests that the bacterium can spread from an infected, newly planted tree to nearby older trees, and this is consistent with what has been reported by Green et al. [8]. It also highlights the need for the early detection tools that can be used routinely for checking asymptomatic trees for the pathogen before they leave the nurseries. In Basel, all trees that were positive for *P. syringae* pv. *aesculi* were planted in the park at least six years ago. No freshly planted horse chestnut trees were present in the park. Therefore, the bacterium could have been introduced with the trees planted in earlier years. Alternatively, the pathogen may have spread from horse chestnut trees planted outside of the park. Direct aerial infection and colonization of mature trees by *P. syringae* pv. *aesculi* is possible, as shown by Steele et al. [13]. The bacterium also has the potential to survive in the soil for up to one year [27], and may be transferred by human activities. Nevertheless, low numbers of *P. syringae* pv. *aesculi* were found in the samples from trees that had been in the park for at least six years, and isolation was not successful. This might be an indication that older trees can resist infection by *P. syringae* pv. *aesculi* and are therefore less affected by the disease, as also suggested by McEvoy et al. [4]. However, McEvoy et al. [4] found reduced infection only for trees older than 80 years.

Our study also shows that the sample methodology influenced the successful detection of *P. syringae* pv. *aesculi*. Higher copy numbers of the bacterium and higher isolation success were obtained from the bark samples compared with the exudate samples. These results go along with the finding from an earlier study showing that less DNA from the target bacterium was isolated from necrotic exudates than from the bark samples [8]. In addition, low isolation success of the bacterium from the exudates was reported earlier by McEvoy et al. [4]. As shown by Tewari and Sharma [28], *P. syringae* pv. *aesculi* can be outcompeted by other fast-growing bacteria, which can lead to such a low isolation success, in particular when samples were taken from older cankers [8,13]. This may have played an important role in the present study, but we could not evaluate this because the age of the cankers on the older trees was unknown.

Isolation of *P. syringae* pv. *aesculi* was only possible in our study when high copy numbers from genomic DNA were found by qPCR. As mentioned above, the age of the canker influences the isolation success [8], and therefore competition from other bacteria may have influenced our data. On the other hand, qPCR possibly detected DNA from non-viable bacterial cells. This was demonstrated in our study by the successful detection of *P. syringae* pv. *aesculi* by qPCR in bark samples inoculated with non-viable bacteria. For other bacterial plant pathogens such as *P. syringae* pv. *tomato* and *Erwinia amylovora*, over-estimation of the pathogen load was also observed because of the inability to distinguish

viable from dead cells when using PCR-based methods [15,16]. In this study, only a slow decrease of the detection rate over time was observed, and high copy numbers of bacterial DNA were still recorded 22 days after inoculation. Storing the samples in the dark may have limited the degradation of DNA from non-viable bacterial cells. Nevertheless, stem samples were not autoclaved before inoculation with non-viable bacteria, and therefore other microorganisms would still be present that could degrade the DNA. After inoculation, the samples were covered with a tape that allowed for air exchange, exposing the DNA to various degrading factors such as enzymatic, chemical, or mechanical degradation [29]. Significant differences in the detectability of non-viable bacteria over time were found between bark samples taken from different seedlings. Variation in the size of the bark samples, as well as in the ratio of bark to wood between the seedlings, may have resulted in the different degradation patterns of the bacterial DNA.

### 5. Conclusions

Our study showed that *P. syringae* pv. *aesculi* was present where bleeding cankers on horse chestnut trees were found. Low isolation success correlated with low gene copy numbers and increasing tree age, indicating a possible tolerance of older trees against this disease. However, the scientific literature also reports a high mortality of mature trees. The observation of the potential infection of older trees through infected, freshly planted horse chestnut trees emphasizes the risk of spreading this disease with nursery plants. Disease-free plant production, early detection of the pathogen, and further monitoring are needed to prevent the bacterium from spreading further and threatening old horse chestnut trees. Overall, qPCR offers an efficient method for the monitoring of *P. syringae* pv. *aesculi* in large numbers of samples. Quantification of the pathogen by qPCR can further be used to investigate disease development, and the processes leading to the recovery and survival of infected trees.

**Supplementary Materials:** The following are available online at https://www.mdpi.com/article/10.3390/f12081062/s1. Table S1: Detection of *Pseudomonas syringae* pv. *aesculi* in bark samples and exudates collected from symptomatic trees. Table S2: Number of target gene copies of *Pseudomonas syringae* pv. *aesculi* for trees inoculated with dead bacterium cells.

**Author Contributions:** Conceptualization, S.S. and J.B.M.; methodology, S.S. and C.S.; validation, S.S. and C.S.; formal analysis, S.S. and C.S.; data curation, S.S.; writing—original draft preparation, S.S.; writing—review and editing, S.S., C.S. and J.B.M.; visualization, S.S.; supervision, S.S.; project administration, S.S. All authors have read and agreed to the published version of the manuscript.

**Funding:** Base funding from the Federal Office for the Environment FOEN.

**Institutional Review Board Statement:** Not applicable.

**Informed Consent Statement:** Not applicable.

**Acknowledgments:** We would like to commemorate Hélène Blauenstein. She supported us energetically and unfortunately can no longer witness this publication. We also thank Quirin Kupper and Esther Jung for their technical assistance, as well as Daniel Rigling, Eckehard Brockerhoff and the reviewers for assistive commenting on the manuscript.

**Conflicts of Interest:** The authors declare no conflict of interest. The funders had no role in the design of the study; in the collection, analyses, or interpretation of data; in the writing of the manuscript; or in the decision to publish the results.

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
