# Peer review of "Detectability of Pseudomonassyringae pv. aesculi from European Horse Chestnut Using Quantitative PCR Compared with Traditional Isolation"

_forests, doi:10.3390/f12081062_

Round 1
Reviewer 1 Report
Pseudomonas syringae pv aesculin monitoring was carried out in three Swiss locations (Zurich, Basel and Klingnau) on symptomatic horse chestnut trees. The paper shows the ability to detect from symptomatic horse chestnut trees, through traditional isolation and Real Time PCR and by comparing the two methods. The lower isolation capacity and the low number of copies of the target gene in RT correlated with the increasing age of the symptomatic horse chestnuts. The detectability of DNA from non-viable Psa cells on wood samples over a period of 22 days was tested.
Comments:
-the paper would be improved by the inclusion in the introduction of a brief description of the symptoms of the pathogen described and of the importance of early diagnosis of the disease even in plants with a low pathogen load
- I suggest you divide the results paragraph into the various sub-chapters described in materials and methods; this greatly facilitates understanding
- why for the identification of the colonies you use the real time with gyrase B gene and not that McEvoy et al., (2016) used for the detection?
-in Discussion or Conclusion it would be useful further investigation on the importance of early diagnosis, even in asymptomatic conditions, which would show the importance of RT even with the problem of detection of non-viable cells. This deepening may contribute to a better interpretation of qPCR data regarding the infection stage helping to evaluate disease progress and the infection risk.
- lines 141, 143, 147, 156,163, 167: Pseudomonas syringae pv aesculin it is not written in italics
Author Response
Dear reviewer
We thank for the detailed comments on this manuscript and your help to improve its quality. Please find below our responses to the points raised. Line numbers in brackets refer to the revised version including markups. Some additional changes were made to make the manuscript clearer and more concise.
The paper would be improved by the inclusion in the introduction of a brief description of the symptoms of the pathogen described and of the importance of early diagnosis of the disease even in plants with a low pathogen load.
We have added a short description of the symptoms at different infection stages of the bacterium Pseudomonas syringae pv. aesculi (L34-37, L44, L87-88). Because of the addition of a new reference, the numbering has been changed accordingly.
I suggest you divide the results paragraph into the various sub-chapters described in materials and methods; this greatly facilitates understanding.
We have divided the results into similar paragraphs as used in material and methods (L224-437). Consequently, some parts of the results needed some restructuring for following the new sub-chapters.
Why for the identification of the colonies you use the real time with gyrase B gene and not that McEvoy et al., (2016) used for the detection?
Many thanks for this comment. The sequencing was done routinely in order to identify bacteria that were cooccurring with Pseudomonas syringae pv. aesculi. Additionally to the sequencing, all colonies were tested by qPCR. This has been corrected accordingly (L150-151).
In Discussion or Conclusion it would be useful further investigation on the importance of early diagnosis, even in asymptomatic conditions, which would show the importance of RT even with the problem of detection of non-viable cells. This deepening may contribute to a better interpretation of qPCR data regarding the infection stage helping to evaluate disease progress and the infection risk.
A short section regarding the importance of early diagnosis has been added (L473-474, L542).
Lines 141, 143, 147, 156,163, 167: Pseudomonas syringae pv aesculin it is not written in italics.
This has been changed accordingly.
Reviewer 2 Report
This paper illustrated the detection of Pseudomonas syringae pv. aesculi on chestnut in Europe using qPCR method. I have one small suggestion for the Table 1, and the number of positive samples for qPCR and Isolation should be listed in two separate row.
Author Response
Dear reviewer
We thank for the comments on this manuscript and your help to improve its quality. Please find below our responses to the points raised. Some additional changes were made to make the manuscript clearer and more concise. English language and style have been checked extensively.
I have one small suggestion for the Table 1, and the number of positive samples for qPCR and Isolation should be listed in two separate rows.
Many thanks for this comment. We reduced the font size for Table 1 to make clear that the results for qPCR and Isolation are presented in two separate columns.
Reviewer 3 Report
The article can be published with minor revision
Author Response
Dear reviewer
We thank you for the comments on this manuscript and your help to improve its quality. Some additional information were added about the symptoms caused by the pathogen as well as the importance of its early detection (L34-37, L44, L86-86, L492-493). Line numbers in brackets refer to the revised version including markups. Some additional changes were made to make the manuscript clearer and more concise.